# Inhibition of p90RSK Ameliorates PDGF-BB-Mediated Phenotypic Change of Vascular Smooth Muscle Cell and Subsequent Hyperplasia of Neointima

**DOI:** 10.3390/ijms24098094

**Published:** 2023-04-30

**Authors:** Ae-Rang Hwang, Hee-Jung Lee, Suji Kim, Seong-Hee Park, Chang-Hoon Woo

**Affiliations:** 1Department of Pharmacology, Yeungnam University College of Medicine, 170 Hyeonchung-ro, Nam-gu, Daegu 42415, Republic of Korea; dofkddofkd@hanmail.net (A.-R.H.); lhjung9351@gmail.com (H.-J.L.); 2Division of Cardiovascular Disease Research, Department of Chronic Disease Convergence Research, Korea National Institute of Health, 187 Osongsaengmyeng 2-ro, Osong-eub, Heungdeok-gu, Cheongju-si 28159, Republic of Korea; ksjyj0105@korea.kr; 3Department of Physiology, Ewha Womans University College of Medicine, 25 Magokdong-ro 2-gil, Seoul 07804, Republic of Korea; 4Senotherpy-Based Metabolic Disease Control Research Center, Yeungnam University College of Medicine, 170 Hyeonchung-ro, Nam-gu, Daegu 42415, Republic of Korea

**Keywords:** p90 ribosomal S6 kinase (p90RSK), platelet-derived growth factor (PDGF), vascular smooth muscle cell (VSMC), neointima formation

## Abstract

Platelet-derived growth factor type BB (PDGF-BB) regulates vascular smooth muscle cell (VSMC) migration and proliferation, which play critical roles in the development of vascular conditions. p90 ribosomal S6 kinase (p90RSK) can regulate various cellular processes through many different target substrates in several cell types, but the regulatory function of p90RSK on PDGF-BB-mediated cell migration and proliferation and subsequent vascular neointima formation has not yet been extensively examined. In this study, we investigated whether p90RSK inhibition protects VSMCs against PDGF-BB-induced cellular phenotypic changes and the molecular mechanisms underlying the effect of p90RSK inhibition on neointimal hyperplasia in vivo. Pretreatment of cultured primary rat VSMCs with FMK or BI-D1870, which are specific inhibitors of p90RSK, suppressed PDGF-BB-induced phenotypic changes, including migration, proliferation, and extracellular matrix accumulation, in VSMCs. Additionally, FMK and BI-D1870 repressed the PDGF-BB-induced upregulation of cyclin D1 and cyclin-dependent kinase-4 expression. Furthermore, p90RSK inhibition hindered the inhibitory effect of PDGF-BB on Cdk inhibitor p27 expression, indicating that p90RSK may induce VSMC proliferation by regulating the G0/G1 phase. Notably, treatment with FMK resulted in attenuation of neointima development in ligated carotid arteries in mice. The findings imply that p90RSK inhibition mitigates the phenotypic switch and neointimal hyperplasia induced by PDGF-BB.

## 1. Introduction

Multiple vascular conditions, including hypertension, atherosclerosis, restenosis following coronary intervention, and vein bypass graft failure, are initiated and aggravated by unregulated vascular smooth muscle cell (VSMC) proliferation in the vasculature [1]. VSMCs are normally exceptionally quiescent and contractile, with higher expression of cellular contractile markers such as smooth muscle myosin heavy chain (SM22α), and α-smooth muscle actin (α-SMA) [2,3]. However, in vascular injury or during inflammatory conditions, VSMCs tend to switch into a synthetic phenotype, in which cells migrate into the intima and cause neointimal hyperplasia exhibiting inflammatory characteristics [2,4]. Furthermore, this phenotypic change in cells is characterized by higher proliferation rates and elevated expressions of extracellular matrix (ECM) proteins and likely contributes to the vascular diseases mentioned above [5,6,7]. As a result, therapeutic strategies targeting VSMC phenotypic switching and their enhanced proliferation and migration could be beneficial for treating these pathological conditions.

Platelet-derived growth factor (PDGF) stimulates VSMC proliferation, which plays a key role in atherosclerosis development and the occurrence of restenosis following angioplasty [8,9,10,11]. Upon vascular injury, activated inflammatory and vascular cells release growth factors, which stimulate VSMCs to convert from a contractile state into a synthetic state [9,12,13,14,15]. For instance, PDGF and corresponding receptor expressions are upregulated in coronary arteries after angioplasty [16,17]. The PDGF family consists of five proteins, namely, PDGF-AA, PDGF-AB, PDGF-BB, PDGF-CC, and PDGF-DD [18], and PDGF-BB is a potent stimulator of VSMC proliferation and migration [19].

p90 ribosomal S6 kinase (p90RSK) is a serine/threonine kinase and member of the S6 ribosomal kinase (RSK) family that acts as a downstream transmitter of extracellular signal-related kinase (ERK) signaling [20]. p90RSK consists of an ERK docking site, two kinase domains, one of which is located at the N-terminus and the other at the C-terminus, and a linker region. Following activation, p90RSK interacts with target proteins to regulate the expression of various genes and proteins and influence cell-cycle progression and cell proliferation [21]. The cell cycle is tightly regulated by cyclins, cell cycle activators, and negative regulators of cell cycles, Rb, p16, p21, and p27. Cyclin D is a major cell cycle-associated cyclin that interacts with cyclin-dependent kinase 2 (Cdk2), Cdk4, Cdk5, and Cdk6, and cell cycle progression requires the accumulation of cyclin D and the Cdk4/6 complex (2,3). p27 negatively regulates the kinase enzyme activities of the cyclin and Cdk complexes through direct binding to the complex. Furthermore, after vascular injury, VSMCs divide in response to mitogens, exiting the G1 phase and entering the S phase.

Metabolic stress caused by chronic hyperglycemia or cellular oxidative stress can activate p90RSK [22,23], and elevated p90RSK activation has been detected in various metabolic conditions, including atherosclerosis and diabetic heart disease [24,25,26,27,28]. Therefore, p90RSK inhibition might be an attractive therapeutic strategy for these pathological conditions. Reversible p90RSK inhibitors targeting the N-terminal kinase domain, such as BI-D1870 and SL0101 [29,30], as well as an irreversible inhibitor, FMK, which targets the C-terminal kinase domain [5], have been developed.

The mechanisms underlying the impact of p90RSK on PDGF-BB-mediated phenotypic switching and intimal hyperplasia of VSMC remain to be investigated. Therefore, we investigated the regulatory effects of p90RSK on the PDGF-BB-induced proliferation and migration of rat primary VSMCs and examined its influence on the development of neointimal hyperplasia following carotid artery ligation in a mouse model.

## 2. Results

### 2.1. p90RSK Is Involved in PDGF-BB-Induced Cell Proliferation in VSMCs

The proliferation of VSMCs might be increased due to synthetic phenotypic changes caused by host factors, including cytokines, growth factors, and environmental factors [4,31,32,33]. In this regard, PDGF-BB has been employed to dedifferentiate VSMCs in this study. p90RSK has two kinase domains (one at the N-terminal end and the other at the C-terminal end). BI-DI870 targets the N-terminal kinase domain and acts as a reversible p90RSK inhibitor, while FMK irreversibly inhibits the C-terminal kinase domain of p90RSK. In that sense, we utilized two p90RSK inhibitors, such as BI-DI870 and FMK, that inhibit the N- and C-terminal kinase domains, respectively. To investigate whether p90RSK affects PDGF-BB-mediated cell proliferation, VSMCs were pre-treated with p90RSK inhibitors FMK and BI-D1870 prior to PDGF-BB treatment, followed by MTT assay, 5-bromo-2′-deoxyuridine (BrdU) incorporation assay, and Western blot analysis or immunofluorescence staining. MTT assay revealed that PDGF-BB induced VSMC proliferation in a dose-dependent manner, and this proliferation was significantly suppressed by FMK or BI-D1870 (Figure 1A–C). Previous studies reported the in vitro IC50 for BI-D1870 and FMK to be around 30 nM; however, the cell-based assay regarding agonist-induced phosphorylation of p90RSK substrates in a cell culture system revealed that the IC50s of BI-D1870 and FMK were approximately 1 μM and 0.2 μM, respectively [5,30]. Therefore, we determined the doses of p90RSK inhibitors at 1, 2, 5, and 10 μM. In addition, similar findings were observed in the cell counting assay (Figure 1D). To examine the effect of p90RSK inhibitors on cell cytotoxicity, apoptosis was determined by Annexin V/PI double-staining. As shown in Figure 1E, FMK (10 μM) and BI-D1870 (10 μM) did not induce apoptotic cell death in the flow cytometric analysis using FITC-conjugated Annexin V. These findings indicate that FMK and BI-D1870 inhibit PDGF-BB-induced cell proliferation without affecting cytotoxicity. To confirm the role of p90RSK in PDGF-BB-mediated cell proliferation, BrdU incorporation assays and Ki-67 expression were conducted. PDGF-BB-induced BrdU incorporation and Ki-67 expression were significantly inhibited by p90RSK inhibitors, respectively (Figure 1F–H). These findings indicate that p90RSK is involved in the regulation of cell proliferation by PDGF-BB in VSMCs.

### 2.2. p90RSK Is Involved in PDGF-BB-Mediated Phenotypic Switching

VSMCs are normally exceptionally quiescent and contractile, with higher expression of cellular contractile markers such as smooth muscle myosin. In this regard, we investigated the functional significance of PDGF-induced activation of p90RSK in the phenotypic switching of VSMCs. The effect of p90RSK on PDGF-BB-induced phenotypic switching was examined in VSMCs by using the chemical inhibitors FMK and BI-D1870 or an adenoviral vector encoding a dominant negative form of p90RSK (Ad-DN-p90RSK). As shown in Figure 2A,B, PDGF-BB-mediated upregulation of collagen I and osteopontin (OPN) was inhibited by FMK and BI-D1870 treatment, but the SM22α level was enhanced in VSMCs (Figure 2A,B). Consistent with the results of the chemical inhibition study, the PDGF-BB-mediated upregulation of collagen I and OPN was significantly inhibited by Ad-DN-p90RSK, but the SM22α level was enhanced (Figure 2C). Since p90RSK isoforms are four, we used DN-p90RSK, which inhibits the p90RSK isoforms via dimerization with endogenous p90RSK isoforms, to exclude the off-target effects of chemicals. In addition, we investigated whether the same results could be obtained by overexpressing p90RSK. Overexpression of p90RSK using an adenoviral vector encoding p90RSK increased the protein expression of OPN induced by PDGF-BB (Figure 2D). Moreover, the mRNA levels of collagen I and III were significantly upregulated by PDGF-BB while suppressing α-SMA and SM22α levels. These alterations of synthetic phenotypic markers were significantly reversed by FMK or BI-D1870 treatments (Figure 2E–H). These results indicate that p90RSK might be a potential regulator of PDGF-BB-mediated VSMC phenotypic switching.

### 2.3. p90RSK Regulates PDGF-BB-Induced Migration of VSMCs

To examine the role of p90RSK in cell migration, VSMCs were treated with FMK or BI-D1870, and PDGF-BB-induced cell migration was determined using the scratch and migration chamber assays (Figure 3A,B). Pretreatment of VSMCs with FMK or BI-D1870 significantly suppressed PDGF-BB-mediated cell migration as determined by the scratch assay (Figure 3A), and this was supported by migration chamber assay results (Figure 3B). In addition, overexpression of p90RSK using an adenoviral vector encoding p90RSK significantly increased cell migration (Figure 3C). These findings indicate that inhibition of p90RSK negatively regulates PDGF-BB-mediated migration of the cells.

### 2.4. p90RSK Inhibition Prevents PDGF-BB-Induced Cell Cycle Progression of Rat Primary VSMCs

To examine the effects of FMK or BI-D1870 on cell cycle progression, levels of cell cycle regulatory molecules were quantified using Western blot analysis and flow cytometry. We found that PDGF-BB increased cyclin D1 and CDK4 protein levels but reduced p27 levels (Figure 4A). On the other hand, FMK or BI-D1870 suppressed cyclin D1 and CDK4 protein levels but enhanced p27 levels in VSMCs (Figure 4A). In addition, the results of flow cytometry revealed that PDGF-BB significantly blocked cells in the G0/G1 phase compared to the control group, but the distribution increased in both the S and G2/M phases. In contrast, FMK and BI-D1870 significantly increased the number of cells in the G0/G1 phase and significantly decreased the number of cells in both the S and G2/M phases compared to PDGF-BB. These findings imply that p90RSK inhibition blocks PDGF-BB-induced VSMC proliferation by regulating cell cycle arrest.

### 2.5. FMK Attenuates Neointimal Hyperplasia Induced by Carotid Artery Ligation

Our previous studies found FMK-mediated p90RSK inhibition and no significant adverse effects from FMK in vivo in mice [25,34]. To clarify the role of p90RSK on cell proliferation and migration in VSMCs, the effect of FMK on the proliferation of smooth muscle cells was determined in a mouse model of carotid artery ligation. In an experimental model of carotid artery ligation, VSMC proliferation is a critical pathological process contributing to the development of neointima. The left common carotid artery (LCA) was ligated, and the effect of FMK on the development of neointima was determined 3 weeks after ligation. As shown in Figure 5A, FMK administration decreased neointima width compared to that of the vehicle control (Figure 5A), but no significant changes were observed in the non-ligated right common carotid artery (RCA) (Figure 5A). The increased expression of collagen was observed in the ligated LCA by comparing it to that of the non-ligated RCA. Furthermore, up-regulated expression of collagen was inhibited by FMK administration (Figure 5B). To better understand the role of the p90RSK in the proliferation of VSMCs in vivo, proliferating cells were assessed by immunohistochemical staining against proliferating cell nuclear antigen (PCNA) on the ligated arteries treated with vehicle or FMK. As shown in Figure 5C, FMK administration decreased PCNA-positive cells compared to those of the ligated LCA (Figure 5C), but no significant changes were observed in the non-ligated right common carotid artery (RCA). These findings imply that p90RSK inhibition prevents VSMC proliferation and subsequent vascular remodeling in a mouse model of carotid artery ligation.

## 3. Discussion

Elevated VSMC migration and proliferation are responsible for restenosis after angioplasty and atherosclerosis [13]. In this study, we observed that p90RSK inhibitors (FMK or BI-D1870) mitigated the PDGF-BB-induced differentiation, migration, and proliferation of rat primary VSMCs. Additionally, FMK ameliorated hyperplasia of the neointima following carotid artery ligation in mice. These results provided insights into the molecular mechanisms through which p90RSK inhibition prevents vascular diseases, particularly regarding the regulation of VSMC proliferation, migration, and phenotypic switching.

VSMC phenotypic switching is a major cellular event during vascular disease progression [35,36]. PDGF-BB-induced activation of p90RSK and PDGF-BB-induced proliferation and migration have been reported in VSMCs; however, a link between p90RSK and PDGF-BB-induced proliferation and migration of VSMCs has not been established. p90RSK is a well-known target of the MAPK signaling cascade for cellular migration and proliferation [37,38]. p90RSK affects various cellular functions, including differentiation and proliferation, in an ERK MAPK-dependent manner [29]. Under inflammatory conditions, elevated PDGF-BB induces VSMC migration through an MAPK-dependent mechanism [39,40]. Early MAPK activation underlies PDGF-BB-mediated VSMC migration, and p90RSK serves as a regulator of MAPK signaling pathways in these cells [31,39]. Our findings indicate that p90RSK can be a selective pharmaceutical target for abnormal VSMC phenotypic switching without affecting the ERK MAPK signaling pathway.

It has been well known that kinase inhibitors usually have off-target effects because of the similarity of the ATP binding site [41]. For example, Stathopoulou and colleagues reported divergent off-target effects of p90RSK inhibitors in cardiomyocytes [42]. They showed that BI-D1870, an ATP-competitive amino-terminal kinase inhibitor of p90RSK, could inhibit phosphodiesterase, resulting in the elevation of the protein kinase A-dependent pathway under conditions of adrenergic receptor stimulation. However, FMK, an allosteric carboxyl-terminal kinase inhibitor of p90RSK, did not show the inhibitory effect of phosphodiesterase, suggesting the therapeutic potential and superiority of FMK for drug development. It was also reported that BI-D1870, but not other p90RSK inhibitors, induces p21 and apoptosis in a p90RSK-independent manner in a colon carcinoma cell line [43]. The authors used a relatively high dosage of BI-D1870 (10 μM), and the regulation of p90RSK expression did not affect BI-D1870-mediated p21 expression. In the current study, we used two different types of p90RSK inhibitors, and both BI-D1870 and FMK showed similar effects on the regulation of phenotypic switching of VSMCs. In addition, we confirmed p90RSK-dependent regulation using the dominant negative form of p90RSK, which inhibits p90RSK isoforms via dimerization of endogenous p90RSK. Considering the development of kinase inhibitors as therapeutics, it is crucial to reduce the off-target effects of novel compounds through allosteric inhibition without affecting the ATP pocket.

In various pathological conditions of the cardiovascular system, such as hypertension and atherosclerosis, vascular inflammation is essential for vascular remodeling [44]. Various host and tissue factors, such as adhesion molecules, pro- and anti-inflammatory cytokines, and growth factors, are involved in the formation of atherosclerotic lesions. In particular, increased expressions of PDGF have been reported in the arterial walls of atherosclerotic regions and in infiltrating inflammatory cells [40]. PDGF is a potent regulator of VSMC proliferation and migration, which ultimately induce restenosis. As a result, the identification of agents capable of blocking PDGF signaling in VSMCs might be useful for treating proliferative vascular diseases.

VSMCs with a synthetic phenotype exhibit accelerated proliferative and migratory abilities that are critical for the development of vascular diseases, including atherosclerosis and restenosis after angioplasty or bypass surgery [3]. Furthermore, PDGF-BB-induced VSMC proliferation and migration are important mechanisms in vascular diseases with abnormal proliferation of VSMC. We therefore sought to elucidate the mechanism underlying PDGF-BB-induced cellular signaling by p90RSK in VSMCs. MTT, cell counting, scratch wound, and migration chamber assays showed that pretreatment with FMK or BI-D1870 reduced PDGF-BB-induced VSMC proliferation and migration, which indicated the involvement of p90RSK in the PDGF-BB-induced proliferation and migration of VSMCs (Figure 1 and Figure 3).

CDK4 and cyclin D1 exert proliferative effects on VSMCs; however, p27 is expected to counter these [2,45]. In addition, cyclin D1 and p27 regulate the progression of the cell cycle with opposing effects on VSMC proliferation [46]. The results of our Western blot and flow cytometric analyses revealed that FMK or BI-D1870 mitigated the upregulation of cyclin D1 and CDK4 protein expressions as well as the downregulation of p27 protein expression in VSMCs incubated with PDGF-BB (Figure 4A). In the G0/G1 phase, the number of PDGF-BB–treated cells significantly decreased compared to that of the control cells, whereas the number of FMK or BI-D1870-treated cells significantly increased compared to that of PDGF-BB-treated cells, suggesting that p90RSK inhibition blocks cell proliferation by regulating cell cycle arrest (Figure 4). Notably, FMK administration significantly suppressed collagen induction and the development of neointima in the ligated left common carotid artery (LCA) in mice (Figure 5). We did not use BI-D1870, a specific inhibitor of p90RSK, in the smooth muscle cell proliferation assay in the mouse carotid artery ligation model because we have more experience with FMK than BI-D1780 in in vivo experiments. FMK did not cause serious adverse effects in animals and also showed similar inhibitory effects against p90RSK as BI-D1870 in in vitro assays. These findings suggest that p90RSK inhibition ameliorates vascular remodeling by preventing the phenotypic switch of VSMCs in vivo.

The ECM of vascular walls acts as a scaffold that provides VSMCs with anchorage and mobility but also delivers mechanical cues to VSMCs that regulate their shapes, metabolism, migration, proliferation, and differentiation [47,48,49]. VSMCs are surrounded by a complex, highly structured ECM consisting of collagen, elastin, and proteoglycans. Matrix molecules play crucial roles in guiding cell function and maintaining tissue structure [50]. To further elucidate the response of VSMCs to growth factors that may be linked to ECM-mediated adhesion and migration, we used in vitro assays, including Western blotting, scratch wound, and migration chamber assays. The biochemical data revealed that FMK and Ad-DN-p90RSK both markedly inhibited the PDGF-BB-mediated upregulation of collagen I and III but enhanced SM22α levels (Figure 2A,B). In addition, PDGF-BB significantly enhanced the mRNA expressions of ECM molecules (collagen I and III) but suppressed α-SMA and SM22α levels. Furthermore, FMK and BI-D1870 significantly reduced the production of ECM molecules (Figure 2C–H). These results indicate that p90RSK inhibition can be a potent therapeutic strategy for not only vascular diseases with abnormal VSMC proliferation but also fibrosis-related diseases with EMT and ECM accumulation. A deeper understanding of these cell-matrix interactions is required to identify novel therapeutic targets and prevent adverse arterial wall remodeling in atherosclerosis and restenosis [51,52].

Overall, the findings of our study revealed that p90RSK participates in regulating PDGF-BB-induced proliferation and migration of VSMCs and expression of ECM, both in vitro in rat primary cultured VSMCs and in vivo in mice. Our findings also suggest that p90RSK inhibition offers a potential strategy for treating vascular diseases associated with aberrant VSMC proliferation and vascular remodeling.

## 4. Materials and Methods

### 4.1. Reagents and Antibodies

MTT reagents were purchased from Amresco (Solon, OH, USA). PDGF-BB was from Cell Signaling (Darmstadt, Germany), and FMK and BI-D1870 were from Axon Medchem (Groningen, Netherlands). Antibodies in this study were as follows: mouse anti-CDK4 (DCS-35, cat. no. sc-23896), mouse anti-OPN (cat. no. sc-73631), mouse anti-Ki-67 (cat. no. SC-23900), mouse anti-PCNA (cat. no. SC-56), and rabbit anti-p27 (C-19, cat. no. SC-528) were supplied by Santa Cruz Biotechnology (Delaware, CA, USA), and rabbit anti-cyclin D1 (92G2, cat. no. #2978) was from Cell Signaling Technology (Danvers, MA, USA), rabbit anti-collagen type I (cat. no. NB600-408) was from NOVUS biologicals (E Easter Ave, CO, USA), rabbit anti-SM22α (cat. no. ab14106) was from Abcam (Cambridge, UK), and mouse anti-tubulin (cat. no. T5168) was from Sigma-Aldrich (St. Louis, MO, USA), and goat anti-mouse IgG-HRP (cat. no. GTX213111-01) and goat anti-rabbit IgG-HRP (cat. no. GTX213110-01) were from GeneTex (Irvine, CA, USA).

### 4.2. Vascular Smooth Muscle Cell Culture

Isolation and culture of primary VSMCs were conducted as follows: aseptically isolated thoracic aorta was chopped into 1 mm-long pieces and cultured in a 100 mm tissue culture dish in DMEM (Welgene Inc., Kyungsan, Korea) containing 50% fetal bovine serum (FBS; Welgene Inc.) and 1% antibiotics-antimycotics at 37 °C in a humidified 5% CO_2_ incubator. The purity of isolated primary VSMC was determined by immunostaining with an anti-SM22α antibody, and more than 95% of cells were positive for SM22α. Rat primary VSMCs were then maintained in DMEM containing 10% FBS and 1% penicillin/streptomycin solution at 37 °C in a humidified 5% CO_2_ incubator. In all experiments, cells from 4 to 7 passages were used.

### 4.3. Animals

All animal experiments were performed at Yeungnam University College of Medicine following the guidelines of the Korean Institutional Animal Care and Use Committee, and all the experimental procedures performed in animals, including the use of anesthetic reagents zoletil, xylazine, and avertin, were reviewed and approved by the Institutional Animal Care and Use Committee of Yeungnam University College of Medicine (Daegu, Korea). Specific pathogen-free C57BL/6 mice (male, 25–30 g, n = 23, 7–8 weeks old) and Sprague-Dawley rats (male, 150–200 g, n = 10, 8 weeks old) were purchased from Samtaco (Seoul, Korea) and maintained in a controlled environment of 22 °C and 40–50% humidity, with 12 h of light per 24 h period, and fed standard rodent chow and water ad libitum peri-operatively. For the isolation of rat primary VSMCs, Sprague-Dawley rats were anesthetized by intraperitoneal (i.p.) injection with a Zoletil (30 mg/kg) and xylazine (10 mg/kg) cocktail; the thoracic aorta was isolated aseptically; and VSMCs were isolated as described in the section on VSMC culture.

Partial ligation of the left LCA was performed in 8-week-old male C57BL6 mice as described in the section on the carotid artery ligation model. At the end of experiments, mice were anesthetized with avertin (500 mg/kg, i.p.), and respiratory failure and death were induced by thoracotomy. Saline was perfused, followed by perfusion with 4% paraformaldehyde. After excision of the left and right carotid arteries, the vessels were immersed in 70% ethanol. For isolation of aortic tissues, mice were euthanized by cervical dislocation under avertin anesthesia (500 mg/kg, i.p.). The aortic tissue and the plasma were stored at −80 °C until analysis. The humane endpoint was set at 20% body weight loss and/or inability to ambulate.

### 4.4. Western Blotting Assay

To extract proteins, cells were lysed with radioimmunoprecipitation assay (RIPA) lysis buffer containing 1 mM phenylmethylsulfonyl fluoride and 0.01 mM protease inhibitor cocktail for 15 min on ice following centrifugation at 15,000× *g* for 15 min at 4 °C. Protein concentration was measured by the Bradford assay. The 30 μg protein samples were then separated by SDS-PAGE with 8–12% gels, transferred to polyvinylidene difluoride (PVDF) membranes, and then incubated overnight with primary antibodies (1:1000) at 4 °C. PVDF membranes were blocked using 5% skimmed milk for 1 h at room temperature prior to primary antibody incubation. Membrane bound antibodies were then detected with corresponding secondary antibodies (1:5000, 1 h at room temperature), and the immunoreactive signal on the membrane was developed with electrochemiluminescence (ECL) reagents (Millipore, Temecula, CA, USA). Western blots were quantified by densitometry after normalization with tubulin, which is a housekeeping gene. In the case of p90RSK phosphorylation, we quantitated the band intensities of phospho-p90RSK and total p90RSK and examined the ratio of phosphorylated p90RSK versus total p90RSK. Results are summarized as a bar graph.

### 4.5. Quantitative Real-Time RT-PCR

Relative mRNA expressions of target genes were assessed by qRT-PCR, as previously described [53]. Total RNAs were isolated with TRIzol^®^ (Invitrogen, Carlsbad, CA, USA), and reverse transcription with 1 μg of total RNA was performed with TaqMan RT reagents (Applied Biosystems Inc., Austin, TX, USA). The qRT-PCR reaction was performed with 1 μL template cDNA mixtures using 2X Power SYBR Green Master Mix in an ABI PRISM 7500 Fast Sequence Detection System (ABI). Relative expressions of target genes were determined by ΔΔCt method, and GAPDH was used as an endogenous control to normalize target gene expression. Expression level of target gene (ΔCt) was normalized by subtracting Ct_GAPDH_ from Ct_Target gene_, and ΔΔCt_Target gene_ of the sample was calculated by subtracting the average ΔCt of control samples from ΔCt_Target gene_ of each sample. Fold gene expression was calculated as 2^−(ΔΔCt). qRT-PCR was performed using a preliminary denaturation at 50 °C for 2 min, followed by 40 amplification cycles of 95 °C for 10 min, 95 °C for 15 s, and 60 °C for 1 min.

The sequences of the specific primers used were as follows: collagen type I: forward, 5′-ATCAGCCCAAACCCCAAGGAGA-3′ and reverse, 5′-CGCAGGAAGGTCAGCTGGATAG-3′; collagen type III: forward, 5′-AGGGAACAACTGATGGTGCTACTG-3′ and reverse, 5′-GGACTGCTGTGCCAAAATAAGAGA-3′; α-SMA: forward, 5′-GCCGAGATCTCACCGACTAC-3′ and reverse, 5′-GTCCAGAGCGACATAGCACA-3′; SM22α: forward, 5′-TGTTCCAGACTGTTGACCTC-3′ and reverse, 5′-GTGATACCTCAAAGCTGTCC-3′; GAPDH: forward, 5′-ACAAGATGGTGAAGGTCGGT-3′ and reverse, 5′- AGCTTCCCATTCTCAGCCTTGA-3′.

### 4.6. MTT Assay

The PDGF-BB-induced proliferation of the cells was quantified by the MTT assay. Cells were cultured in a 24-well tissue culture plate, and the cell culture medium was changed with fresh DMEM upon reaching approximately 80% cell confluency. After that, cells were pretreated with 2 or 5 μM of BI-D1870 or FMK and stimulated with PDGF-BB (1, 2, 5, or 10 ng/mL) for 24 h. Cells were treated with MTT reagents and incubated for an additional 4 h at 37 °C in the dark. Plates were centrifuged, washed with PBS, and crystal precipitates were dissolved with DMSO. The optical density at 570 nm was read using a microplate reader (Bio-Rad, Hercules, CA, USA).

### 4.7. BrdU Incorporation Assay

Cells were seeded in 96-well tissue culture plates at a density of 5 × 10^3^ cells/well. Prior to PDGF-BB treatment, it was pretreated with the p90RSK inhibitors FMK and BI-D1870 for 1 h, followed by DNA synthesis measured by 5-bromo-2′-deoxyuridine (BrdU) incorporation 24 h after PDGF-BB treatment. Briefly, after 24 h of incubation with PDGF-BB, the BrdU labeling solution was added to the cells and incubated for 24 h. Media was removed from the cells, and then DNA was denatured by adding 100 μL of fixation/denaturation solution to each well. Subsequently, a horseradish peroxidase-labeled anti-BrdU monoclonal antibody was added, and the plate was incubated at room temperature for 90 min. The BrdU–antibody complexes were detected by a colorimetric reaction with the 100 µL TMB substrate. PDGF-BB-induced cell proliferation was confirmed using an assay kit (Novus Biologicals, Colorado, CO, USA), and optical density was measured at 450 nm in a microplate reader (Bio-Rad).

### 4.8. Adenoviral Transduction

To examine the effect of p90RSK on PDGF-BB-induced phenotypic switching of VSMCs, adenoviral vectors encoding LacZ (Ad-LacZ), wild-type p90RSK (Ad-WT-p90RSK), or the dominant negative form of p90RSK (Ad-DN-p90RSK) were used. Adenoviral vectors were amplified and tittered in AD293 cells by using the Adeno-XTM Rapid Titer Kit (Clontech Laboratories, Mountain View, CA, USA), following the manufacturer’s instructions. VSMCs were transduced with adenoviruses at 2 or 10 of the multiplicity of infection (MOI) for 1 day in a growth medium and then treated with PDGF-BB (2 ng/mL) for 24 h.

### 4.9. Ki-67 Immunofluorescent Staining

Cells treated with PDGF-BB in the presence or absence of FMK and BI-D1870 pretreatment were fixed with 10% formalin for 1 h and permeabilized for 10 min at room temperature. Cells were then blocked with 5% normal goat serum in PBS with 0.1% Tween-20 and incubated overnight at 4 °C with a monoclonal anti-Ki-67 antibody (1:200). After washing 3 times with PBS, the samples were incubated with the secondary antibody (1:2000) (Alexa Fluor^TM^ 546 goat anti-mouse IgG, cat. no. A-11003; Invitrogen, Carlsbad, CA, USA) and DAPI for 1 h at room temperature. Ultimately, the signals were observed using an immunofluorescence microscope (Olympus).

### 4.10. Flow Cytometric Assay

For cell-cycle analysis, trypsinized cells were fixed in 95% ethanol, followed by staining with 50 μg/mL propidium iodide (PI) for 30 min at 37 °C. Cells were then filtered using a 5 mL polystyrene round-bottom tube fitted with a cell-strainer cap. Flow cytometry was performed using a BD FACSCalibur (Becton Dickinson, San Jose, CA, USA), and the acquisition was analyzed by CellQuest Pro (Beckton Dickinson).

### 4.11. Annexin V-FITC Apoptosis

The Annexin V-FITC apoptosis detection kit I (BD-Bioscience, Franklin, NJ, USA) was used to detect apoptosis according to the instructions of the manufacturer. First, VSMCs were cultured with PDGF in the presence or absence of FMK and BI-D1870 for 24 h, and the cells were collected into glass tubes. Next, the cells were stained with Annexin V solution at 37 °C for 15 min, followed by staining with PI at 37 °C for 15 min. Ultimately, the cells were quantified using flow cytometry using CellQuest Pro (Beckton Dickinson). Based on flow cytometry analysis, normal cells, early and late apoptotic cells, and necrotic cells were present in the lower left, lower right, upper right, and upper left quadrants, respectively.

### 4.12. Cell Migration Assay

For the wound scratch assay, rat primary VSMCs in a six-well tissue culture plate were cultured until 90% cell confluency was reached. The medium of the plates was then changed with fresh serum-free DMEM and incubated overnight. A wound scratch was made by drawing a line through cells perpendicular to the bottom of the well with an autoclaved 200 μL pipet tip. Scratched cells were pretreated with BI-D1870 (2, 5 μM) or FMK (2, 5 μM), followed by 24 h of treatment with 2 ng/mL of PDGF-BB in the presence of Ara C, a specific inhibitor of cell proliferation. We examined each group’s images at different time points, comparing images between the initial scratch images (0 h) and after PDGF treatment at two different time points (12 h and 24 h). The images of the cells were obtained under phase-contrast microscopy at 40 x magnification, and cell migration was quantified as a percentage of the original scratch area. For the cell migration chamber assay, VSMCs grown in an inner chamber of a trans-well tissue culture plate (0.3 μm pore size) were pretreated with BI-D1870 (2, 5 μM) or FMK (2, 5 μM), and the lower chamber was filled with 2 ng/mL of PDGF-BB. At the end of the incubation, cells were fixed and stained with crystal violet. The migrated cells stained purple were enumerated under a microscope.

### 4.13. Carotid Artery Ligation Model

Eight-week-old, specific pathogen-free, C57BL/6 male mice were obtained from Samtaco (Seoul, Korea). For the carotid artery ligation, mice were anesthetized by i.p. injection of avertin (15 mg/kg), and the left common carotid artery (LCA) was exposed by blunt dissection. Three of the four caudal branches of the LCA, which are the left external carotid, internal carotid, and occipital arteries, were ligated using a 6–0 silk suture as described previously [54]. After surgery, mice were treated with FMK (5 mg/kg, i.p.) every other day for 3 weeks. At the end of experiments, animals were anesthetized and perfused with PBS, followed by perfusion with 4% paraformaldehyde. Both the left and right carotid arteries were excised and embedded in paraffin. Lumen diameter, lumen area, neointima area, media area, and total vessel area were measured using NIH Image J (National Institutes of Health, Bethesda, MD, USA).

### 4.14. Histology

The cross-sections of ligated LCA were analyzed at predefined proximal distances from the ligation site. The section distance to the bifurcation of internal and external carotid arteries was determined in the non-ligated right common carotid artery (RCA). Paraffin-embedded tissues were sectioned at 5 μm thickness, deparaffinized with xylene, dehydrated, and stained with hematoxylin and eosin. The elastic laminarity in the vessels was visualized by autofluorescence under a confocal microscope (Leica, Bannockburn, IL, USA).

### 4.15. Immunohistochemistry

Aorta tissues were fixed in 10% buffered formalin, paraffin-embedded, and cut into 5 μm slices. Sections were subjected to immunohistochemistry for PCNA expression. Sections were deparaffinized, rehydrated, blocked, and then incubated with mouse anti-PCNA antibody (1:200) (cat. no. sc-56, Santa Cruz Biotechnology, Delaware, CA, USA) overnight at 4 °C. The immunohistochemistry was performed using the HRP/DAP (ABC) detection IHC kit (Abcam, Cambridge, UK), following the manufacturer’s instructions. All the microscopic images of the sections were obtained using an optical microscope (Nikon, Tokyo, Japan).

### 4.16. Statistical Analysis

The data in bar graphs are presented as mean ± SD, and the significance of intergroup differences was assessed by unpaired Student’s *t*-test and multiple group comparisons using ANOVA followed by Bonferroni’s *post hoc* test, with *p* values < 0.05 indicative of a significant difference. The analysis was conducted using GraphPad Prism (Graph-Pad Software Inc.).

## Figures and Tables

**Figure 1 ijms-24-08094-f001:**
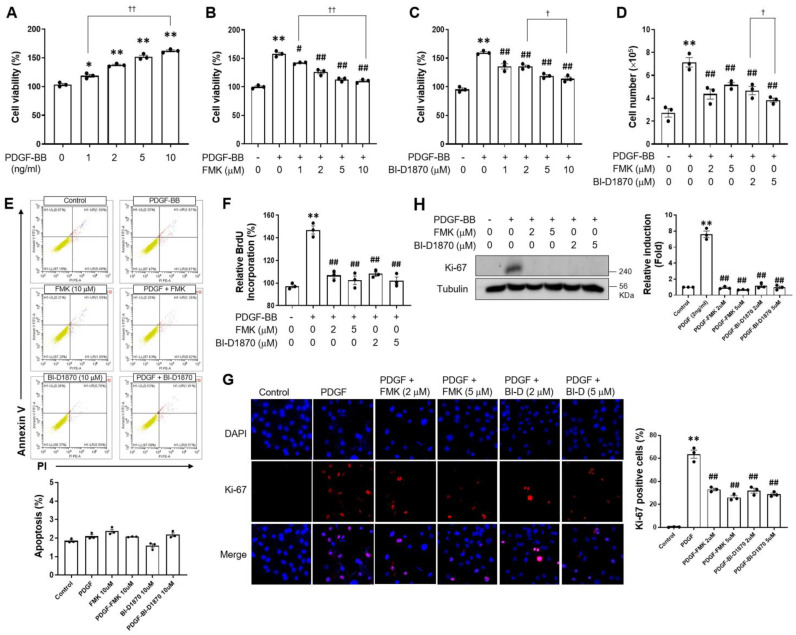
Inhibition of p90RSK ameliorates PDGF-BB-induced proliferation of rat primary VSMCs. (**A**) VSMCs cultured in serum-free media were treated for 24 h with various concentrations of PDGF-BB as indicated. * *p* < 0.05, ** *p* < 0.01 vs. vehicle control (lane 1), ^††^
*p* < 0.01. (**B**,**C**) VSMCs cultured in serum-free media were treated for 1 h with FMK (1, 2, 5, 10 μM) or BI-D1870 (1, 2, 5, 10 μM), followed by treatment with 2 ng/mL of PDGF-BB for 24 h. Following incubation, the viability of the cells was assessed by MTT assay. ** *p* < 0.01 vs. vehicle control (lane 1), # *p* < 0.05, ## *p* < 0.01 vs. cells treated with PDGF-BB (line 2), ^†^
*p* < 0.05, ^††^
*p* < 0.01. (**D**–**H**): cell counting (**D**), BrdU incorporation (**F**), immunofluorescence (**G**), and Western blot analysis (**H**) were performed to determine the proliferation of the cells induced by PDGF-BB. ** *p* < 0.01 vs. vehicle control (lane 1), ## *p* < 0.01 vs. cells treated with PDGF-BB (line 2), ^†^
*p* < 0.05. The rate of apoptosis was determined by flow cytometry using the Annexin V-FITC/PI staining assay (**E**). (i) The lower left quadrant represents the viable cells; (ii) the lower right quadrant shows the early apoptosis cells; (iii) the upper right quadrant signifies the late apoptosis cells; and (iv) the upper left quadrant indicates dead cells. The data are representative of the three independent experiments.

**Figure 2 ijms-24-08094-f002:**
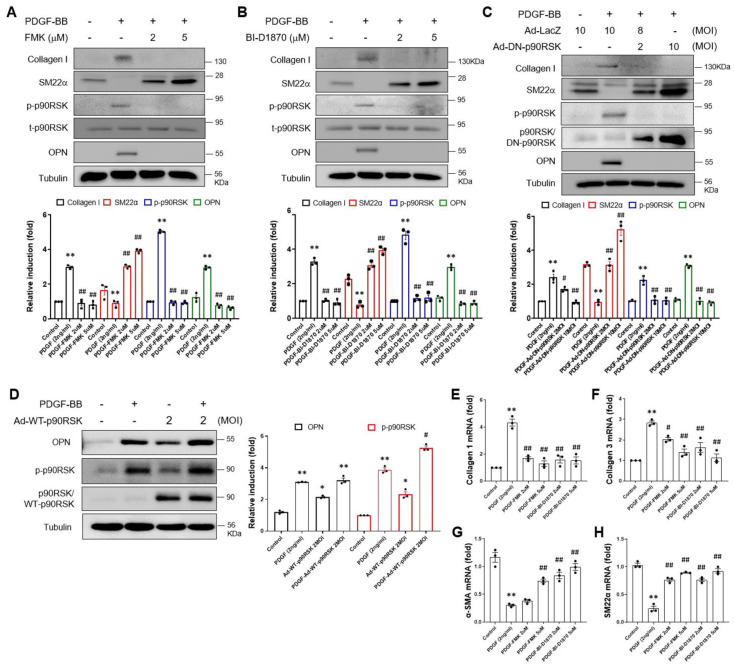
p90RSK is involved in PDGF-BB-mediated phenotypic switching of VSMCs. (**A**,**B**) VSMCs were treated for 1 h with FMK and BI-D1870, followed by treatment with 2 ng/mL of PDGF-BB for 24 h. The expression level of collagen I, SM22α, p90RSK, and OPN proteins was determined via immunoblotting. The results of the densitometric analysis of immunoblotting bands were presented in the bar graphs. ** *p* < 0.01 vs. vehicle control (lane 1), ## *p* < 0.01 vs. cells treated with PDGF-BB (lane 2). (**C**,**D**) VSMCs were transduced either with Ad-WT-RSK and Ad-DN-RSK or Ad-LacZ as a control for 24 h, followed by treatment with 2 ng/mL of PDGF-BB for 24 h. The level of protein expression was analyzed via immunoblotting. * *p* < 0.05, ** *p* < 0.01 vs. vehicle control (lane 1), # *p* < 0.05, ## *p* < 0.01 vs. cells treated with PDGF-BB (lane 2). (**E**–**H**) The mRNA expression level of collagen I, collagen III, α-SMA, and SM22α was determined by quantitative RT-PCR assay in VSMCs treated with FMK or BI-D1870. VSMC was pretreated with either 2 or 5 μM of FMK or BI-D1870 for 1 h, followed by treatment with 2 ng/mL of PDGF-BB for 24 h. The data are representative of the three independent experiments. ** *p* < 0.01 vs. vehicle control (lane 1), # *p* < 0.05, ## *p* < 0.01 vs. cells treated with PDGF-BB (lane 2).

**Figure 3 ijms-24-08094-f003:**
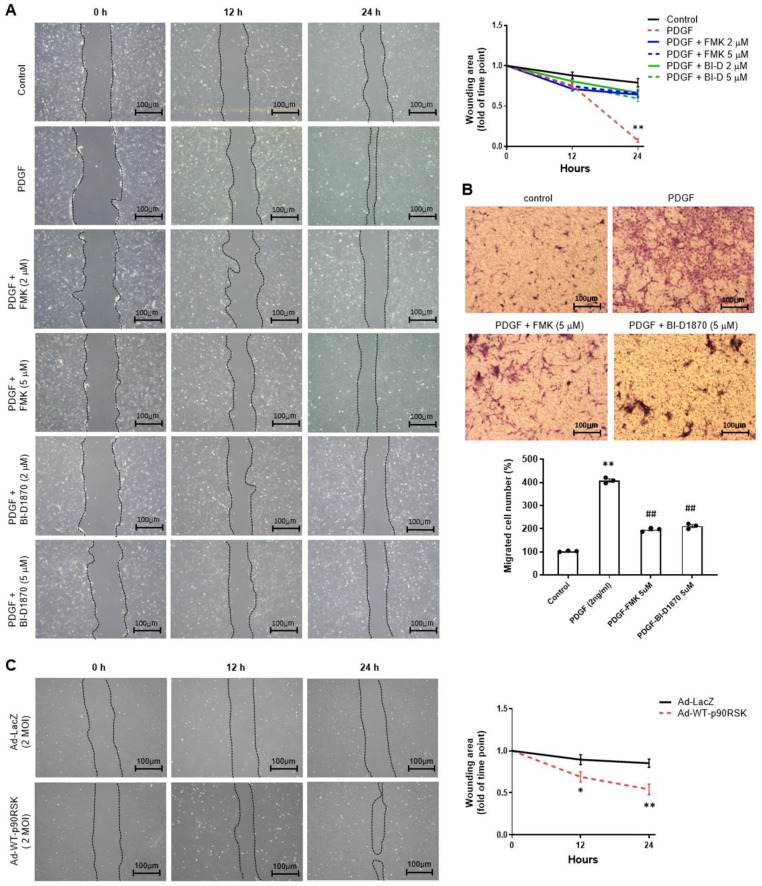
p90RSK regulates PDGF-BB-induced cell migration of VSMCs. (**A**) Serum starved VSMCs grown near confluent were wounded and then treated for 1 h with either 2 or 5 μM of FMK or BI-D1870, followed by treatment with 2 ng/mL of PDGF-BB for 24 h. The wound area was observed at different time points as indicated and recorded under a phase-contrast microscope. ** *p* < 0.01 vs. cells treated with PDGF-BB. (**B**) The Transwell system was used for the selective migration assay. VSMCs were pretreated for 1 h with various concentrations of either FMK or BI-D1870, followed by treatment with 2 ng/mL of PDGF-BB for 24 h. The number of migrated cells to the lower chamber was enumerated under the light microscope. ** *p* < 0.01 vs. vehicle control (lane 1), ## *p* < 0.01 vs. cells treated with PDGF-BB (lane 2). (**C**) VSMCs grown near confluent were transduced either with Ad-WT-p90RSK or Ad-LacZ as a control for 24 h, and then serum-starved cells were wounded. The wound area was observed at different time points as indicated and recorded under a phase-contrast microscope. The data are representative of the three independent experiments. Scale bar, 100 μm. * *p* < 0.05, ** *p* < 0.01 vs. vehicle control (Ad-LacZ).

**Figure 4 ijms-24-08094-f004:**
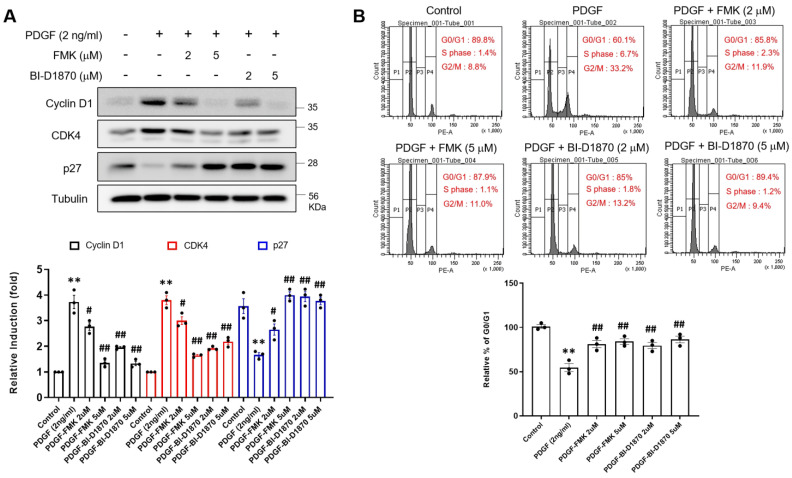
p90RSK regulates PDGF-BB-induced cell cycle progression in VSMCs. (**A**,**B**) Serum-starved VSMCs were pretreated for 1 h with either 2 or 5 μM of FMK or BI-D1870, followed by treatment with 2 ng/mL of PDGF-BB for 24 h. (**A**) The expression level of cyclin D1, CDK4, p27, and tubulin protein was determined by immunoblotting assay. The bar graphs illustrate the results of the densitometric analysis of the immunoblotting bands. ** *p* < 0.01 vs. vehicle control (lane 1), # *p* < 0.05, ## *p* < 0.01 vs. cells treated with PDGF-BB (lane 2). (**B**) For the cell cycle analysis, VSMCs grown in a six-well tissue culture plate, were pretreated for 1 h with either 2 or 5 μM of FMK or BI-D1870, followed by treatment with 2 ng/mL of PDGF-BB for 24 h. To determine the cell cycle, collected cells were analyzed by flow cytometric assay. ** *p* < 0.01 vs. vehicle control (lane 1), ## *p* < 0.01 vs. cells treated with PDGF-BB (lane 2).

**Figure 5 ijms-24-08094-f005:**
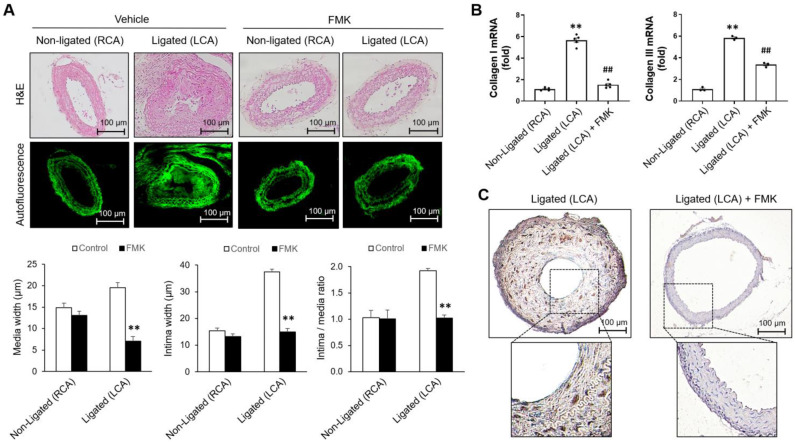
Neointimal hyperplasia was inhibited by FMK in a ligated carotid artery in mice. In a mouse model of partial carotid ligation, animals were i.p. inoculated for 3 weeks with FMK (5 mg/kg per day, every other day). (**A**) H&E stained arteries of mice (at 200× magnification, top panel). Elastic laminae were visualized by autofluorescence using a confocal microscope. Graphical measurement of the media and intima (n = 4; mean ± SE, bottom panel). Scale bar, 100 μm. H&E, hematoxylin, and eosin stain. ** *p* < 0.01 vs. ligated (LCA, lane 1). (**B**) The arteries isolated from the mice were analyzed for the mRNA expression of collagen I and collagen III by a quantitative RT-PCR assay. The data are representative of the three independent experiments. ** *p* < 0.01 vs. non-ligated (RCA, lane 1), ## *p* < 0.01 vs. ligated (LCA, lane 2). LCA, left common carotid artery; RCA, right common carotid artery (**C**) PCNA expression in aorta tissue sections from a mouse model of carotid artery ligation-induced neointimal hyperplasia was determined by immunohistochemistry with an anti-PCNA antibody. Scale bar, 100 μm.

## Data Availability

The data are contained within the article.

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
