# Peer review of "Inhibition of p90RSK Ameliorates PDGF-BB-Mediated Phenotypic Change of Vascular Smooth Muscle Cell and Subsequent Hyperplasia of Neointima"

_ijms, 2023, doi:10.3390/ijms24098094_

Round 1
Author Response
Response to Reviewer 1 Comments
The revised manuscript with all changes (marked in RED) is attached.
Point 1: When the authors selected specific inhibitors of p90RSK, FMK and BI-DI870 inhibitors, what was the reason for the choice, whether there were differences between the two inhibitors or what adverse reactions did they have to VSMC cells?
Response 1: It was known that p90RSK has two kinase domains (one at the N-terminal end and the other at the C-terminal end). BI-DI870 targets the N-terminal kinase domain and acts as a reversible p90RSK inhibitor, while FMK inhibits the C-terminal kinase domain and irreversibly inhibits p90RSK. In that sense, we utilized two p90RSK inhibitors such as BI-DI870 and FMK that inhibit N- and C-terminal kinase domain, respectively. Here we examined the dose-dependent responses of both p90RSK inhibitors in cell viability and apoptosis assay and no significant decreases in cell viability or cytotoxicity were observed in the VSMC cells (Figure 1). We described this point in the section of Result in the revised manuscript.
Point 2: In Figure 1, the authors used the MTT method to explore the effect of p90RSK inhibitors on PDGF-BB-mediated cell proliferation. The authors then determined the dose of PDGF-BB and two p90RSK inhibitors by MTT method. Could the authors further explain the question of the dosage for VSMC cells and C57 mouse species?
Response 2: Although in vitro IC50 is around 30 nM as reported in the original articles for BI-D1870 and FMK, the cell-based assay regarding agonist-induced phosphorylation of p90RSK substrates in cell culture system showed that IC50s of BI-D1870 and FMK were approximately 1 mM and 0.2 mM, respectively. Therefore, we determined the doses of p90RSK inhibitors at 1, 2, 5, and 10 mM. We described this point in the section of Result in the revised manuscript.
Point 3: The author proposed in Result 2 that there are 4 subtypes of p90RSK, but the DN-p90RSK used by the author selectively inhibits the p90RSK subtype, can the author describe in depth why the DN-p90RSK subtype was chosen in the discussion section so that the reader can better understand your topic.
Response 3: The 90kDa ribosomal S6 kinases (RSKs) are a serine/threonine kinase group consisting of four RSK isoforms (RSK1-4). DN-p90RSK, which inhibits the p90RSK isoforms via dimerization with endogenous p90RSK isoforms, was used to exclude the off-target effects of chemicals. Although it is common to knock down p90RSK using a siRNA system, here we used the DN-p90RSK because it is difficult to inhibit all four isoforms with siRNA system in primary VSMCs. We described this point in the revised manuscript.
Point 4: In the results Figure 5A and 5C, the immunofluorescence and immunohistochemistry results section, can the author add the correct ruler to all the pictures, and adjust the size of the pictures to the same size.
Response 4: As suggested by the reviewer, we carefully checked the ruler to the pictures and adjusted the size of pictures in the results for immunofluorescence and immunohistochemistry analysis.
Point 5: The author does not seem to be very effective in proving your conclusion in the entire results section: “Inhibition of p90RSK ameliorates PDGF-BB-mediated phenotypic change of vascular smooth muscle cell and subsequent hyperplasia of neointima”. The authors added two specific inhibitors of p90RSK to explore the morphological and protein changes of VSMC cells, but whether the authors can add knocking down p90RSK on VSMC cells or increase the expression of p90RSK to further verify whether the same results are achieved, which will appear that the author's discussion is more rigorous and accurate.
Response 5: As suggested by the reviewer, we examined whether the same results were obtained by p90RSK overexpression. Overexpression of p90RSK using adenoviral vector encoding p90RSK caused an increase in protein expression of osteopontin (OPN) induced by PDGF-BB (Fig 2D). In addition, we found that cell migration was significantly increased by p90RSK overexpression (Fig 3C). The original data have been replaced with these data in the revised manuscript. As explained in response for Point 3, we did not try knocking down of p90RSK because it is difficult to inhibit all four isoforms with siRNA system in primary VSMCs.
Point 6: The author has some deficiencies in the layout of the results, whether the size of the grouping font size on the picture can be adjusted to be consistent, as shown in Figure 3B.
Response 6: As suggested by the reviewer, the grouping font size was adjusted to be consistent, as shown in Figure 3B, and the original data were replaced in the revised manuscript.

Reviewer 2 Report
In this study Hwang et al, show that p90RSK participates in the regulation of PDGF-BB-induced proliferation and migration of vascular smooth muscle cells and extracellular matrix components in vitro in rat primary cultured vascular smooth muscle cells and also in vivo in mice. The authors show that the inhibition of p90RSK using FMK and BI-D187 can offer a potential strategy for the treatment of vascular diseases associated with impaired vascular smooth cell proliferation and vascular remodeling.
Major
1. Figure 1 A, B, C and D, if there is an effect in dose dependent manner, in this case the data should be analyzed by comparing several groups to each other.
2. In Figure 2, the authors used, an adenoviral vector 125 encoding a dominant negative form of p90RSK (Ad-DN-p90RSK)., this study also need an empty vector as a control.
3. In Figure 4, p90RSK regulates PDGF-BB-induced cell cycle progression in VSMCs
4. The authors presented G0/G1 phase but not G2M and S phase. This data should be also included.
5. In Figure 4, relative induction and G0/G1 fold presented. It is not clear which group was compared with which one. Please provide this information, also for all other data in the manuscript. Indicate with lines, comparisons between groups, samples.
6. It is also necessary to discuss in more details, how a therapeutically approach could be possible. Please discuss the potential off targets of use of FMK and BI-D1870. (PMID: 24136223, PMID: 31344438) they can also influence cell cycle independently of p90RSK, through p21/p53, the regulators of cell cycle
Minor
1. Please delete following template sentence from the introduction.
The introduction should briefly place the study in a broad context and highlight why it is important.
The proliferation and migration of VSMCs might be increased due to synthetic phenotypic changes caused by host factors, including cytokines, growth factors, and en vironmental factors [2-5]. VSMCs are normally exceptionally quiescent and contractile with higher expression of cellular contractile markers, such as smooth muscle myosin
2. This sentence should be in the result part. In this regard, PDGF-BB has been employed to dedifferentiate VSMCs in this study.
Author Response
Response to Reviewer 2 Comments
The revised manuscript with all changes (marked in RED) is attached.
Point 1: Figure 1 A, B, C and D, if there is an effect in dose dependent manner, in this case the data should be analyzed by comparing several groups to each other.
Response 1: We agree reviewer’s comment and analyzed statistical differences by comparing different dosage groups to each other. To show the effects of dose responses, statistical differences between middle dosage and high dosage were analyzed. There are significant statistical differences indicating the dose responses of PDGF and p90RSK inhibitors in cell proliferation. We added new statistics in the revised manuscript (Figure 1).
Point 2: In Figure 2, the authors used, an adenoviral vector encoding a dominant negative form of p90RSK (Ad-DN-p90RSK). This study also needs an empty vector as a control.
Response 2: We are sorry for inconvenience regarding missing label in detail. Actually, we used Ad-LacZ as a control adenoviral vector in the experiment and this information has been added in the revised manuscript.
- In Figure 4, p90RSK regulates PDGF-BB-induced cell cycle progression in VSMCs. The authors presented G0/G1 phase but not G2M and S phase. This data should be also included. In Figure 4, relative induction and G0/G1 fold presented. It is not clear which group was compared with which one. Please provide this information, also for all other data in the manuscript. Indicate with lines, comparisons between groups, samples.
Response 3: As suggested by the reviewer, we added G2/M and S phase in the new Figure 4. The results of Flow cytometry showed that PDGF-BB significantly blocked cells in the G0/G1 phase compared to the control group, but the distribution increased in both S and G2/M phases. In contrast, FMK and BI-D1870 significantly increased the number of cells in the G0/G1 phase and significantly decreased the number of cells in both the S and G2/M phases compared to PDGF-BB. These results suggest that p90RSK inhibition prevents PDGF-BB-induced cell cycle progression.
In order to clearly indicate statistical significances, each group comparisons were assigned a lane number, and these details were described in the revised Figure legends like as *p<0.05 vs. vehicle control (lane 1) or ##p<0.01 vs. cells treated with PDGF-BB (lane 2).
Point 4: It is also necessary to discuss in more details, how a therapeutically approach could be possible. Please discuss the potential off-targets of use of FMK and BI-D1870. (PMID: 24136223, PMID: 31344438) they can also influence cell cycle independently of p90RSK, through p21/p53, the regulators of cell cycle.
Response 4: Thank you very much for reviewer’s valuable comments regarding off-target effects of p90RSK inhibitors. It has been well known that kinase inhibitors usually have off-target effects because of the similarity of ATP binding site (REF: ACS Chem Biol. 2015 Jan 16;10(1):234-45. doi: 10.1021/cb500886n). For example, Stathopoulou and colleagues reported divergent off-target effects of p90RSK inhibitors in cardiomyocytes (REF: PMID: 31344438). They showed that BI-D1870, an ATP-competitive amino-terminal kinase inhibitor of p90RSK, could inhibit phosphodiesterase resulting elevation of PKA-dependent pathway under conditions of adrenergic receptor stimulation. However, FMK, an allosteric carboxyl terminal kinase inhibitor of p90RSK, did not show the inhibitory effect of phosphodiesterase suggesting therapeutic potential and superiority of FMK for drug development. It was also reported that BI-D1870, but not other p90RSK inhibitors, induces p21 and apoptosis via the p90RSK-independent manner in a colon carcinoma cell line (REF: PMID: 24136223). The authors used a relatively high dosage of BI-D1870 (10 mM) and the regulation of p90RSK expression did not affect BI-D1870-mediated p21 expression. In the current study, we used two different types of p90RSK inhibitors and both BI-D1870 and FMK showed similar effects on the regulation of phenotypic switching of VSMCs. In addition, we confirmed p90RSK-dependent regulation using dominant negative form of p90RSK which inhibits p90RSK via dimerization of endogenous p90RSK. Considering the development of kinase inhibitors as therapeutics, it is crucially important for reducing off-target effects of novel compounds in a way of allosteric inhibition without affecting the ATP pocket.
Point 5: Please delete following template sentence from the introduction. ‘The introduction should briefly place the study in a broad context and highlight why it is important.’
Response 5: We deleted that template sentence from the introduction in the revised manuscript.
Point 6: ‘The proliferation and migration of VSMCs might be increased due to synthetic phenotypic changes caused by host factors, including cytokines, growth factors, and environmental factors. VSMCs are normally exceptionally quiescent and contractile with higher expression of cellular contractile markers, such as smooth muscle myosin.’ This sentence should be in the result part. In this regard, PDGF-BB has been employed to dedifferentiate VSMCs in this study.
Response 6: We agree reviewer’s comment and moved this sentence in the section of Result. In addition, we mentioned that PDGF-BB has been employed to dedifferentiate VSMCs into synthetic phenotypic changes in the revised manuscript.

Round 2
Reviewer 1 Report
The authors have investigated on “Inhibition of p90RSK ameliorates PDGF-BB-mediated phenotypic 2 change of vascular smooth muscle cell and subsequent hyper3 plasia of neointima ”. The study may useful to increase in this field. However, several points are not clear and need to be addressed as follows;
1. The grammar of the abstract part is confused. It is suggested to revise and polish it and reorganize the word order.
2. Part of the figure DAPI in Figure 1G is not the same image as the Merge, and the Ki-67 location is different from the one on the Merge;It is recommended to add quantitative analysis。
3. Figure 1E should be flush with other figures
4. Note analysis confusion, for example, Figure 1A,B,C note in the back, D,F,G,H note in the front.
5. Please add the WB strip to the corresponding molecular weight. Figure 2C, please replace the cluttered strip。
6. When shooting the scratch experiment, try to ensure that the same aperture, so that the exposure state is the same, avoid taking pictures of different shades.
7. Figure 5. What is the ruler size?If you put the ruler on the chart, put it all on the chart. Don't put part on the chart, part in the chart note. The group drawing is too rough, some pictures are not arranged neatly.
8. The use of "*P" in the statistical analysis in Figure 5A is easy to be confused, so please use it in a standard way.
9. The figure legend should not be a detailed description of the experimental process, it should be a simple description of the content of the figure, annotation and summary.
10. The authors of the discussion section should have refined the conclusions of the previous paper by adding deeper mechanistic aspects to the study, rather than spending a lot of time repeating the experimental results.
Author Response
Response to Reviewer Comments
The revised manuscript with all changes (marked in RED) is attached.
Point 1: The grammar of the abstract part is confused. It is suggested to revise and polish it and reorganize the word order.
Response 1: As suggested by the reviewer, the entire manuscript including the abstract was requested to an English proofreading company (editage; editage.com) to be corrected. The certificate was attached and the editing can be seen in the version of track-change.
Point 2: Part of the figure DAPI in Figure 1G is not the same image as the Merge, and the Ki-67 location is different from the one on the Mergeï¼›It is recommended to add quantitative analysis.
Response 2: We are sorry for any inconvenience regarding a mismatch image. Now we checked the all images and corrected. In addition, expression level of Ki-67 was quantified and presented as bar graph in the new Figure 1G. PDGF-BB-induced Ki-67 expression was significantly inhibited by FMK and BI-D1870. We described this in the revised manuscript.
Point 3: Figure 1E should be flush with other figures.
Response 3: As suggested by the reviewer, we set the height of Figure 1E to be the same as the other pictures in the new Figure 1.
Point 4: Note analysis confusion, for example, Figure 1A, B, C note in the back, D, F, G, H note in the front.
Response 4: As suggested by the reviewer, the notes in the figure were unified by putting them on the front side in the revised manuscript.
Point 5: Please add the WB strip to the corresponding molecular weight. Figure 2C, please replace the cluttered strip.
Response 5: As suggested by the reviewer, we indicated the molecular size marker and replaced the cluttered strip in Figure 2C (p-p90RSK) in the revised manuscript.
Point 6: When shooting the scratch experiment, try to ensure that the same aperture, so that the exposure state is the same, avoid taking pictures of different shades.
Response 6: As suggested by the reviewer, we calibrated the background shading to be similar in the new Figure 3.
Point 7: Figure 5. What is the ruler size? If you put the ruler on the chart, put it all on the chart. Don't put part on the chart, part in the chart note. The group drawing is too rough, some pictures are not arranged neatly.
Response 7: As suggested by the reviewer, we marked scale bars in all pictures in Figure 5 and modified group labels in detail in the new Figure 5.
Point 8: The use of "*P" in the statistical analysis in Figure 5A is easy to be confused, so please use it in a standard way.
Response 8: As suggested by the reviewer, we used **p in the statistical analysis in new Figure 5A.
Point 9: The figure legend should not be a detailed description of the experimental process, it should be a simple description of the content of the figure, annotation and summary.
Response 9: As suggested by the reviewer, we tried to simplify the description of the figure legend in the revised manuscript.
Point 10: The authors of the discussion section should have refined the conclusions of the previous paper by adding deeper mechanistic aspects to the study, rather than spending a lot of time repeating the experimental results.
Response 10: As suggested by the reviewer, we added discussions of mechanistic aspects to the study in the section of Discussion in the revised manuscript.
